# A Qualitative Study on Young Men’s Experiences of Intentional Weight-Gain

**DOI:** 10.3390/ijerph20043320

**Published:** 2023-02-14

**Authors:** Craig Donnachie, Helen Sweeting, Kate Hunt

**Affiliations:** 1School of Social and Political Sciences, University of Glasgow, Glasgow G12 8RS, UK; 2School of Psychological Sciences and Health, University of Strathclyde, Glasgow G1 1QE, UK; 3Retired—Previously MRC/CSO Social and Public Health Sciences Unit, School of Health and Wellbeing, University of Glasgow, Glasgow G3 7HR, UK; 4Institute for Social Marketing and Health, University of Stirling, Stirling FK9 4LA, UK

**Keywords:** body weight, male body image, food consumption, eating, masculinity

## Abstract

This qualitative study investigated how young men perceive their body image and experiences of purposively gaining weight, and what these reveal about broader sociocultural meanings around food, consumption and male body image. The participants in this study were a subsample of men participating in the ‘GlasVEGAS’ study which examined the effect of weight-gain and weight loss on metabolism, fitness and disease risk in young adult men. Twenty-three qualitative, semi-structured interviews were conducted with thirteen men (mean age 23 years) at GlasVEGAS baseline (*n* = 10) and weight-gain (6-week) follow-up assessment (*n* = 13). Data were analysed using the principles of framework analysis. The majority of men viewed the foods provided as part of the GlasVEGAS study as ‘luxury’ items despite their being of low nutritional value. The weight-gain process prompted men to reflect on how cultural norms and social environments may amplify overeating. Several described being surprised at how quickly they assimilated unhealthy eating habits and/or gained weight. Some valued changes in their appearance associated with weight-gain, including appearing larger or having increased muscle size. These factors are vital to consider when developing weight management initiatives targeting young men, including the valorisation of unhealthy foods, wider social influences on diet and male body image ideals.

## 1. Introduction

Predominant Western sociocultural constructions of masculinity may pose challenges for men in relation to managing their weight [1], and, particularly for young men, body image. This in turn may influence their willingness to engage in lifestyle behaviours vital for promoting current and future health and wellbeing. However, men remain largely underrepresented in food and eating-related research [2,3].

Men’s food choices may be influenced by dominant ideals of masculinity [4]. For example, it has been suggested that consumption of high fat foods, red meat and excessive amounts of alcohol, can be powerful ways of constructing masculine identities [5,6,7]. It has also been argued that men are more likely to appraise foods through satiety and palate as opposed to their health or nutritional benefits [8]. Moreover, men can regard weight loss efforts, such as dieting and slimming, as ‘feminine’ activities [9] and young adult men in particular may underestimate their weight status when classified overweight or obese [10]. However, there is a dearth of research investigating how young men experience changes in their weight status during young adulthood, typically defined as aged 18–35 years.

Over recent decades it has been suggested that idealised notions of masculinity and body image have been reformulated in Western culture, epitomised by a muscular physique or body build. While strength has historically been linked to physical labour and masculinity, within an increasingly consumerist culture, men’s bodies are valued more by how they appear [11]. Such ideals are portrayed via the media, where male bodies have become more visible [12]. Consistent with the notion that gendered ideals around masculinities and body image oscillate, it has been suggested men are more likely to desire *increased muscle mass* congruent with the ‘mesomorphic’ (highly muscular and lean) build [13]. However, some men may accept an increase in overall body size including increased fat, especially if losing body fat is perceived as difficult.

Findings suggest that men classed as both ‘underweight’ and ‘overweight’/‘obese’ are more likely to experience body dissatisfaction and weight stigma than those classified as ‘normal’ weight [14]. However, it is important to highlight concerns raised over current biomedical conceptions of overweight and obesity [15]. For instance, the Body Mass Index (BMI) does not differentiate between lean body tissue (muscle mass) and adipose tissue (body fat). Thus, individuals with elevated muscle mass may have a high BMI but low body fat [16]. While body image and related pressures, namely the ‘drive for thinness’ [17] amongst women have received significant attention, men remain understudied in body image research [18]. Male body dissatisfaction has been associated with negative psychological consequences, such as depression, decreased self-esteem and, for some, a form of body dysmorphic disorder (‘muscle dysmorphia’) whereby individuals perceive themselves as inadequately muscular and/or large [19]. Research suggests this is mainly experienced by young men, adversely impacting their lives [19,20,21]. Some men with body dissatisfaction may feel compelled to resort to extreme pursuits to align themselves with prevailing sociocultural ideals of the masculine body (e.g., excessive exercise, dysfunctional eating and abuse of appearance- and performance-enhancing drugs) which can result in adverse health issues [22,23]. Moreover, men tend to express eating disorder symptoms differently to women [24] and may delay help-seeking due to cultural norms which frame these as predominantly ‘female issues’ [25].

Recent research suggests many young men experience body image dissatisfaction throughout their everyday lives, extending beyond muscularity and/or leanness, exemplified by growing concerns about appearance [26,27]. However, they may be reluctant to disclose these concerns [28,29] because to do so would be to lose ‘masculine capital’ [5,30,31]. Thus, while young men are expected to tacitly work towards attaining male (Western) body ideals, they must simultaneously appear indifferent to, or detached from, their bodily appearance, denoted as ‘the double-bind of masculinity’ [32]. 

It is against this background that we present a longitudinal qualitative investigation of the experiences of young men, conducted within the unusual and novel context of a clinical study which sought to compare the effect of weight-gain and weight loss on metabolism, fitness and disease risk in young adult South Asian and European men (Glasgow Visceral and Ectopic Fat With Weight Gain in South Asians study—‘GlasVEGAS’: https://clinicaltrials.gov/ct2/show/NCT02399423 (accessed on 2 December 2022); https://theses.gla.ac.uk/41083/ (accessed on 2 December 2022)). The research reported here aimed to: understand young men’s general perceptions of male body image; men’s reasons for choosing to participate in the GlasVEGAS study; their expectations, experiences and consequences of weight-gain during the GlasVEGAS study, and explore what these reveal about broader sociocultural meanings around food, consumption and male body image.

## 2. Materials and Methods

### 2.1. Participants and Data Collection

The participants in this qualitative study were sampled from Caucasian European men living in urban Scotland who had previously consented to participate in the GlasVEGAS study. The GlasVEGAS study, conducted by researchers at the Institute of Cardiovascular & Medical Sciences, University of Glasgow, was designed to investigate: whether there are ethnic group differences in adipose (fat) tissue storage in Caucasian European and South Asian men; fat cell function; and metabolic risk factors in response to weight-gain and loss. Eligible participants in the GlasVEGAS study were aged 18–45 years, had a BMI < 27 kg/m², had been weight stable (±2 kg) for >6 months, were either of white European (self-report of both parents of white European origin) or South Asian (self-report of both parents of Indian, Pakistani, Bangladeshi or Sri Lankan origin) ethnic origin (although it is important to recognize that people of South Asian ancestry in the UK may have dual nationalities/identities or consider themselves solely as British regardless of their generational status), and gave informed consent to eating a diet high in sugar and fat to increase their baseline body mass by ~7% (minimum of 5%) over a six week ‘weight-gain phase’, followed by a 12 week ‘weight loss phase’ during which they were supported to lose any weight gained during the experiment. Exclusion criteria for the GlasVEGAS study included diabetes (physician diagnosed or HbA1c ≥ 6.5% on screening), history of cardiovascular disease, regular participation in vigorous physical activity, current smoking, taking drugs or supplements thought to affect carbohydrate or lipid metabolism, or other significant illness. Men received weekly food as part of the GlasVEGAS weight-gain phase (premium ice cream, chocolate bars, potato crisps, cheese, dried fruit and nuts, sugary drinks) to supplement their usual food intake by ~1500–2000 kcal/day and were offered GBP 400.00 to take part in the study. Further details of the GlasVEGAS study are available here: https://clinicaltrials.gov/ct2/show/NCT02399423 (accessed on 2 December 2022). Ethical approval for the GlasVEGAS study was obtained from the College of Medical, Veterinary and Life Sciences Ethics Committee, University of Glasgow (200140035). 

The researchers on the wider GlasVEGAS study were also interested in understanding young men’s experiences of rapid weight gain and subsequent weight loss and therefore approached our research team to undertake an independent qualitative study with a subsample of GlasVEGAS participants. Men enrolled in the study were asked by one of the GlasVEGAS project researchers if they would consider taking part in our qualitative study. The men were given information on the qualitative research which invited them to take part in face-to-face, semi-structured longitudinal interviews examining young men’s body image and their experiences of weight change. When men attending the laboratory for the GlasVEGAS assessments (at baseline and/or 6-week follow-up) indicated that they were willing to consider taking part in our qualitative study, the GlasVEGAS team notified the first author (CD) who was them able to answer any questions they had about our additional research.

In total, 21 Caucasian European and 14 South Asian men participated in the wider GlasVEGAS project with recruitment continuing for 18-months. Due to issues related to the timing of recruiting South Asian men to the wider GlasVEGAS project, we made a pragmatic decision to exclude South Asian men from our qualitative study. Thus, our sample included Caucasian European men only, reflecting those who had already been enrolled on the main GlasVEGAS study during our fieldwork window. Thirteen men (mean age 23 years) of Caucasian European descent who took part in 23 interviews are included in the qualitative analysis (Table 1). Ten men completed interviews at baseline and weight-gain follow-up assessment and three were interviewed only during the weight-gain follow-up assessment as they had already completed their GlasVEGAS baseline assessment before the independent qualitative element of the study commenced. These three men were asked to respond retrospectively to ‘baseline’ questions; no discrepancies were identified through systematic comparison of their accounts with those of the other men.

Participants in the independent GlasVEGAS qualitative study provided written informed consent for their interviews to be recorded digitally and transcribed. All interviews were conducted by CD between September 2015 and March 2016 in a private room within the University of Glasgow and lasted between 30 and 75 min. The baseline interview topic guide included open-ended questions and probes to explore men’s: motivations for taking part in the GlasVEGAS study; perceptions of their body build/weight; and expectations in anticipation of weight-gain. The follow-up interview topic guide included questions to explore any practical, physical and emotional consequences, and/or challenges experienced during weight-gain (see online Appendix A). Verbatim transcription was undertaken by a professional transcription company.

### 2.2. Data Analysis

Data analysis followed the five main stages recommended by thematic framework methodology [33,34]. Firstly, *familiarisation*—each transcript was read, checked against the audio-recording and anonymised (by CD) through the use of pseudonyms and deletion of any identifying information. Secondly, *identifying a thematic framework*—selected transcripts were read by all authors who met repeatedly to discuss the data and agree an analytical or thematic coding framework. Thirdly, *indexing*—the thematic framework was utilised by CD to code the data. Fourthly, *charting*: the data were charted and summarised into the framework matrices by CD, enabling systematic interrogation of the data. Finally, *mapping and interpretation*—all authors interpreted the data deductively, based on the research objectives, and inductively, based on the concepts emerging as analysis unfolded, applying higher level concepts and overarching themes across the data, both cross-sectionally and longitudinally [33,35]. NVivo software (QSR International Pty Ltd. Version 11. QSR International: Burlington, MA, USA, 2015) was used to facilitate data management and retrieval.

## 3. Results

To address the aims of the qualitative study, this paper focuses on the most salient themes identified from our analyses: General perceptions of male body image; reasons for choosing to participate in the GlasVEGAS study and expectations of the weight-gain process; and experiences of weight-gain and consequences of overeating during the GlasVEGAS study: Follow-up assessment. 

### 3.1. General Perceptions of Male Body Image

Despite being within the medically defined threshold for ‘normal’ healthy weight at baseline (i.e., BMI: 18.5–24.9), several men expressed dissatisfaction with aspects of their bodies, either explicitly or implicitly, and particularly a desire to be ‘more muscly’ and/or ‘leaner’ (i.e., less body fat), reflecting common Western sociocultural ideals in relation to masculinity and body image for young adult men: “*I’m trying to go to the gym more. I’d like to be more muscular, fitter. I think most guys would. But yes it’s, not something I’m concerned about.*” (Malcolm, age 20) 

Most acknowledged that prevailing Western sociocultural body ideals for both men and women are recurrently shaped by popular media, which predominantly target women, encouraging unrealistic beauty standards:

*I think with the kind of advent of kind of magazines and lots of the media influences, especially the pressure on girls is, to an extent, it’s creating a body image which is unachievable. And stuff like Photoshop and stuff has got a role to play in that. I guess the same for men, but I just don’t think they [men] are invested so much in, like, magazines and things like that. I mean, there are a few magazines which are popular with men. Not to the same extent, sort of thing*.(Luke, 22)

However, some recognised a gradual shift in the ways that male bodies have been portrayed within mainstream media (e.g., Television, Films and Magazines), predominantly valorising particular physical attributes (e.g., muscularity and low body fat), amplified in recent years by the ubiquity of image-focused social media platforms (e.g., Instagram and YouTube), and associated sport supplement use: 


*There’s always been like the big movie stars […] Arnold Schwarzenegger for example, I mean he’s a big guy and like was always like something I think men were aware of and trying to work towards, but now I think there’s more awareness of it in terms of like the fitness magazines and stuff […] and through social media online and all that sort of stuff […] more publicity I suppose, so, you’re seeing a lot more pictures of guys which are bigger and so it’s becoming more of a regular thing to sort of, gym a lot, the protein shakes and all that sort of stuff to try and get on the extra weight and size and so I think yeah, I think it has increased a lot in the last, last wee while anyway.*
(Douglas, 19)

While several men did not want to be viewed by others as ‘overweight’, most perceived their body build (or shape) to be more important than their actual weight: “*I would probably put more of a focus on how I looked, as opposed to how much I weigh—especially as a guy*” (Philip, 28). Additionally, some men articulated unease or discomfort in relation to being perceived as ‘too thin’: “*I think that I would probably not feel as confident if I […] was too thin or if my build was that way*” (Jack, 20). Some expressed an explicit desire to increase their body build or size and reported previous deliberate attempts to gain ‘healthy weight’ (i.e., muscle mass) congruent with these ideals. For example, some spoke about the challenges they had experienced in prior attempts to gain weight and maintain weight gain before they signed up to take part in the GlasVEGAS study:


*I think, yeah, at the end of the summer I was gaining weight but I was normally gaining, like, healthy weight. I was going to the gym, and drinking protein shakes, and eating more and fatter foods. So, well, I used to weigh, like, 62, 63 [kilograms], probably what I do now [at GlasVEGAS baseline]. So, I went up to 65, 66 at some point but then the studies [university work as a student] kicked in and, yeah, it was lots, like, a lot of work to do, like, different projects etc. So, then it just dropped down.*
(Simon, 21)

The men articulated several reasons underpinning their desire for increased body (muscle) mass, particularly enhanced corporeal abilities (e.g., athletic/physical performance) which they associated with health, and sometimes also improved attractiveness: 


*I’d probably be lying if I didn’t say it was mostly aesthetic. But the health, the health part of it is… I mean, I like to do a lot of outdoor sports as well, so a lot of it’s functional as well. Like, I’d like to be able to climb better and stuff like that.*
(Malcolm, 20)


*Most guys wanna be muscular […] biologically it’s just—well, you wanna be attractive because you want to procreate and therefore you wanna look attractive so that the female is attracted to you and therefore you can procreate. And obviously being muscular shows that you are a better hunter or whatever and they’re, then you’re, kinda, that’s how I imagine that it, kinda, works.*
(Jack, 20)

Some men identified negative consequences from the pressures or effort needed to conform to these bodily ideals. For instance, Marcus (22) expressed feeling more ‘confident’ when he perceived himself as having additional (lean muscle) mass, specifically when he had participated regularly in rugby: “*When I was bigger [increased muscle size], I felt like, I don’t know, maybe a wee bit more confident, maybe, wi’ [with] people you just met*”, while Jack (20) expressed feeling ‘*worse*’ about himself compared to ‘*muscular*’, men: “*Around […] muscular guys for some reason then you’ll feel a little bit worse about yourself and then when you’re around, well, less muscular guys […] you’ll feel […] better about yourself*.”

However, not all participants endorsed these ideals to the same extent. For example, Andy (30) rejected the notion of attempting to enhance his body build, and reflected on his observations of men within commercial gyms and perceived cultural norms around men’s bodies: 

*I think it’s quite important to men, for some reason. I think they feel the bigger they are, it’s a determination o’ [of] their, like their masculinity, for some reason. I don’t perceive that, because I’m quite satisfied in the way I am and the way I look, but I do see it in the industry I currently work in [personal training]. You see it a lot—people putting in a lot o’ [of] effort into getting bigger, doing weird and wonderful things to achieve these goals, and like I wouldn’t be doing that because it’s dangerous, sorta thing*.

Andy (30), the oldest man in the sample, reflected the views of a few of the men who described feeling some pressure to ascribe to particular ‘masculine’ body ideals when they were younger (e.g., during adolescence), but no longer felt the need to embody these physical ideals to the same extent. Some described such practices as ‘superficial’ and/or felt that it was not worth their investment (i.e., time/discipline), preferring instead to focus on other interests and pursuits: 


*I slightly feel more comfortable than… not having to feel muscly or trying to improve my body as much as maybe I did when I was seventeen or eighteen […].*
(Luke, 22)

### 3.2. The GlasVEGAS Study: Reasons for Choosing to Participate in the GlasVEGAS Study and Expectations of the Weight-Gain Process

#### 3.2.1. Reasons for Taking Part in the GlasVEGAS Study

The men gave a variety of reasons for deciding to take part in the study. For several, the financial incentive (GPB 400.00) offered for completion of the study was one of the main motivations for signing-up. For example, Luke (22) said: “*Initially interested, I guess, we’re getting paid to do it and it’s £400*”. Some participants described having a personal interest in aspects of GlasVEGAS’ experimental design, including access to a comprehensive ‘health check’ (body composition, fitness and blood tests etc): “*So, it’s interesting […] to find out things that I wouldn’t be able to find out otherwise, like blood glucose levels and cholesterol levels*” (Robert, 22).

#### 3.2.2. Expectations of the Weight-Gain Process

Most indicated that, prior to taking part in the GlasVEGAS weight-gain stage, they felt particularly excited about being able to eat ‘treats’ or ‘luxury’ (i.e., highly processed/calorie dense) foods and were confident they would be able to reach their weight target: 


*I’ll be expected to eat more […] sweet and savoury stuff, like Ben and Jerry’s […] chocolate and the crisps and stuff like that, so I’ll just be slotting them into the gaps in between meals […] quite a lot of snacking […] there’s definitely an emphasis on the, what you’d originally, sort of, think of as a bit luxury items and stuff like that, and stuff you try and keep away from, so it’s quite good to get a bit of a free lease on it and just get to go all out on ‘pigging out’.*
(Douglas, 19)

Only a few expected that consuming increased processed or unhealthy foods would be challenging because it was not something they usually did, or they had minimal experience of monitoring their food intake.

#### 3.2.3. Prior Experiences of Changes in Weight

Some men described previous periods in their lives when they had experienced changes in their weight, and therefore anticipated that gaining weight would be relatively straightforward: 


*I thought it would be quite easy considering I’d been up at that kind of weight beforehand. OK, it had been a few years, but I thought, ‘I should be able to get to that weight again. It shouldn’t be too hard.’*
(Marcus, 22)

Men generally attributed previous weight changes to specific life events, such as moving out of their family home, transitioning to tertiary education, travelling, or stressful circumstances including studies/coursework, physical injury or ill-health. For instance, Luke (22) described how difficulties adjusting to preparing meals for himself when he went to university had resulted in weight-gain: “*I was also cooking for myself for the first time in my life, so I wasn’t really certain how to prepare it, healthier […] change of diet basically*”. Consequently, some discussed strategies they had used to manage their weight in the past, including calorie restriction and/or increased exercise:


*I went to university for the first time, and because I could eat what I like, I gained weight quite a bit. Then I noticed what was happening and stopped eating, apart from things like toast—and then dropped very quickly. My weight yo-yos quite a lot, but always within a certain range. For the last year or so, it’s been between thirteen and fourteen [stones]—and that’s usually a result of eating a lot and then starving a lot, which I recognise is very unhealthy.*
(Kevin, 28)

#### 3.2.4. Expectations of the Consequences of Weight-Gain

Most men anticipated that reaching the target increase in body fat (~7% of baseline body weight) as per protocol for the GlasVEGAS study, would be ‘easy’ or ‘straightforward’. Several perceived the required level of increase to be small or minimal, thus expecting it would have limited impact on them physically or otherwise. For example, Matt (21) said: “*I told [my male friends] I’m just gaining like four kilos for me… which is pretty much nothing*”, while Philip (28) suggested that: “*I know my BMI is fairly low as well—I could probably manage to put on a fair bit of weight without a huge amount of notice*”. 

Some perceived the opportunity to gain and then lose weight as a personal ‘challenge’ or ‘experiment’. For instance, a few men said they had struggled to gain weight previously and were eager to experience what it was like to actively gain weight: “*I’m quite skinny, so I tend to not put on very much weight at all, and I have quite a high metabolic rate*”. (Malcolm, 20)


*I thought it would be quite fun. The idea of gaining weight, it’s not something I’ve ever done before. … I’ve always actually struggled to put weight on, so I thought it would be quite an interesting thing to do, to kind of gain weight and lose it again.*
(Philip, 28)

Some, especially those who said they had found it challenging to gain weight in the past, even anticipated some positive benefits associated with gaining ‘a small amount’ of weight, such as appearing ‘healthier’: 


*A small amount at seven percent. I don’t think it’ll make huge, drastic changes, no. I actually thought it might make me look a bit healthier and a wee bit more fuller, ‘cause I do know that I have, like, I’m more ectomorph [naturally lean and tall build] on the scale—so maybe a wee bit o’ [of] extra timber [body weight] about the place will make me look a wee bit healthier.*
(Andy, 30)

Some men anticipated that there would be at least some changes in their appearance, including increased body fat around their stomach or torso, particularly those who had experienced weight-gain previously: “*I kind o’ [of] guessed it would be going […] on my stomach. […] Kind o’ [of] man tits*” (Kieran, 24).

Although several men conveyed an aversion towards being considered ‘overweight’ or ‘fat’, most expected any potential changes in their body composition, energy levels, mood and/or confidence levels due to weight-gain as part of the study to be only ‘temporary’: 


*If I start to notice myself looking fat, I think I will be slightly self-conscious. Though I guess in my mind’s eye, I know, hopefully, it’ll just be, you know, a temporary change. So that, I think that will kind of help me justify it and kind of not let it, kind of, affect me too much […].*
(Luke, 22)

Some men said they had deliberately told others that they were taking part in the GlasVEGAS study in order to pre-empt and justify any noticeable changes in their appearance and/or eating habits:


*I always want to, like, appear fit and healthy to other people as well so I suppose I’ve been deliberately letting people know that I’m doing this, so they don’t just sort of see the weight change over the next four weeks and then sort of be thinking to themselves, ‘Oh, he’s put on a bit of, bit of weight,’ so like, at least that way they’ll be aware of it […].*
(Douglas, 19)


*I’ve spoken to a couple of people at uni[versity] about it, and yeah, a lot of people […] they see it as a fun idea to kind of, you know, basically be paid and be given food to gain weight.*
(Philip, 28)

### 3.3. Experiences of Weight-Gain and Consequences of Overeating during the GlasVEGAS Study: Follow-Up Assessment

#### 3.3.1. Experience of Weight-Gain Process

In contrast to their expectations of the weight-gain phase of the GlasVEGAS study, when interviewed at their follow-up assessment most men said that gaining weight was more challenging than anticipated. For instance, some said they were ‘surprised’ by the volume of food they had to consume to gain weight: 


*I was always surprised at how much food it was […] I thought I’d just need two tubs. But six or seven is required. You know, and like six or seven tubs of ice cream [per week], like, that’s, you know, like, it’s an awful lot. […] I didn’t realise this is how much it took to, you know, gain [weight]. ‘Cause it was like stuff like, you know, two kilograms of peanuts, tins of Pringles. […] and I was like, ‘Oh, god, like what have I signed myself up for?’*
(Kieran, 24)

Several said that the ‘novelty’ of eating unrestricted energy dense/processed foods quickly dissipated; they went on to describe it as something they ‘had to do’, using language such as ‘fighting’, ‘battling’ or ‘powering through’: 


*To start off wi’ [with], it was a novelty, eating treats every day. First couple of weeks I gained [weight] quite nicely but recently it’s just been a nightmare. […] I was putting food into my mouth and going ‘Oh I’ve had enough’ but I’ve still had, like, half a can o’ Pringles to go […] I sat here the last time [baseline interview] and says it was gonna be really easy, I’ll do it no problem, and I thought about that throughout the time as well and thinking ‘No, this is really, really hard.’*
(Andy, 30)

Some men described others’ reactions to their attempts to gain weight over such a short period, through increased food consumption, as a source of humour. For example, several described, friends or close family members seeing their experiences of overeating and weight-gain as a ’laugh’ or ‘game’: “*They [flatmates] were just like ‘Oh, Davie’s eating again’ an’, like, stuff like that. But I think they just found it funny*” (Davie, 19). 

Many of the men remarked upon how other people perceived the process as being ‘fun’ or an enviable pursuit, especially peers or other (male) friends: “*My friend likes sweets a lot so when he saw that huge pile on my desk with all the sweets and crisps he just—he was like ‘Oh, so good!’*” (Simon, 21).

The process led some to reflect on how specific cultural norms in daily life promote overeating, particularly social situations involving family or friends. For example, Philip described the social challenges of restrained food consumption: 


*Mothers, they love to feed. Whereas […] saying ‘Actually, can you cook me something that’s really low fat? And I’m losing weight.’ That’s […] more intrusive, and might be more difficult. Whereas this was like ‘Yeah, just pile it on.’ And ‘Here, here, have some chocolate! Have a second dessert!’ […] almost as a game.*
(Philip, 28)

A few reported that others, especially family members, had expressed concern for them about making a deliberate, albeit temporary, attempt to gain weight or alter their body shape: “*They [parents] were also like ‘Yeah, it’s dangerous’ or something like that. ‘Experiments with your body’ and so on*” (Matt, 21).

Most men discussed strategies to ensure they were able to meet the energy surplus requirements for weight-gain, for instance, reducing their activity levels and/or planning when and where they would have access to foods throughout the day. This sometimes included concealing their foods from other people so as not to appear ‘greedy’ and/or expose others to their overeating behaviour, especially family members or partners: 


*I’d say one of the sort of more annoying impacts was when I’d be in the library and I’d literally be eating, you know, out of a jar of Nutella with a spoon and I know people would be walking past me, like, judging me for it an’ that was kind of uncomfortable. Like, I wanted to tell everyone ‘I have to, this is a study, I’m not just incredibly greedy.’*
(Malcolm, 22)

Several men said that they became aware of the number of calories and the amount of food they had to consume, so were able to monitor intake daily. This, in turn, ensured they remained on track to reach their weight ‘targets’:


*You sort of got it into your head […] how many calories were in a, for example, a tub of Pringles, or something like that. It was around nine hundred or something like that. So, you’d be sitting there thinking, ‘All right I need to have—throughout a day I’ll have that and, maybe a bag of Haribo [sweets] or, something like that [crisps] as well. And, that will be enough to make up for the fifteen hundred calories’. So, you’d almost, like […] set targets I guess at the start of the day.*
(Douglas, 19)

Importantly, several also described being surprised at how easy it was to develop unhealthy habits, which, in turn, promoted increased consumption, especially of foods high in fat and/or sugar:


*It kinda gives me a more understanding of like how people get in the habit of, [be]cause when you eat junk food one day it just seems like you want to eat junk food again. And I noticed when I would have pizzas, it was much easier to go and get a bag of crisps rather than if I was having my chicken, sweetcorn […] so I can see how it’s kinda a downward spiral there.*
(Marcus, 22)

Although the majority said they managed to gain at least some weight relatively quickly, some men found prolonged weight-gain more difficult, particularly if they could not sustain their eating patterns (e.g., due to illness). Several reported struggling with weight-gain because they experienced adverse consequences, including discomfort, bloating, nausea or other digestive issues as a result of eating increased amounts of (highly processed) food:


*It’s the chewing and it’s the constant repetition of doing it every day. […] [it] became so tiring and then, like, you’re feeling bloated and you’re not feeling good […] wi’ [with] me being quite a mentally strong person, being vulnerable and, for what I perceived as weak […] ‘This is meant to be easy, this should be easy, it’s only a thousand calories, it’s no’ [not] that much, it’s only a box o’ [of] Pringles’.*
(Andy, 30)

Some men discussed resorting to extreme methods of food consumption to ensure they reached their weight targets, particularly as they reached the final stages of the weight-gain phase:


*I’d get a blender, a whole tub of Ben and Jerry’s, maybe four, five tablespoons of peanut butter, full fat milk, sort of Nutella, and then just blend it and drink it.*
(Malcolm, 20)


*There were a couple of days where I was literally, I couldn’t be bothered thinking about everything that I had to eat. So, I would just stick a tub of Ben and Jerry’s in the microwave, melt it down and then just end up, like, sort of drinking it over an hour or two period.*
(Douglas, 19)

#### 3.3.2. Consequences of the Weight-Gain Process

A few men reported experiencing minimal or no changes in themselves or their physical appearance, as a result of weight-gain, with any small physical changes only visible to themselves or their partners (e.g., when they removed items of clothing around their abdomen) but not to others. A few were ‘surprised’ that the changes were not as apparent as they had anticipated and even reported some positive aspects, such as appearing ‘healthier’, ‘bigger’, or ‘more muscular’: 


*I would have thought that it would show more, like that, I was expecting to, well, just looking in the mirror, to have changed more. […] ‘cause you’re thinking ‘Okay, I only put on, like, three kilos, so, it’s normal that I look pretty much the same.’ But actually now that I know that I gained about five and a half kilos I’m pretty surprised that, like, I’m just looking pretty much exactly the same as I did before the weight-gain. […] I guess I just kinda look a little bit bigger but not even in a bad way. You could almost think that I’m more muscular although it’s all fat, which is weird.*
(Jack, 20)

Some of these perceptions were reinforced by comments from other people, particularly family members: 


*My mum does say she noticed, like, a difference in my face, but like a positive one, kind o’ [of] like I didn’t have enough weight on it before. So she was kind o’ like happy about this.*
(Davie, 19)

However, others perceived considerable adverse changes physically and/or psychologically which they found ‘surprising’, including increased body fat, decreased fitness, lethargy and problems sleeping:


*Psychologically I just […] believe I will feel that I’m more attractive when I’m thinner […] I become out of breath much more quickly, I get tired a lot. I notice that I’m fatter than I used to be and I noticed it in places that I didn’t really expect. I looked down in the shower, one day, and noticed I have fat ankles now. That is what I am. I am a person with fat ankles, and that was not a pleasant realisation at all. So yeah, participating, doing things was fun—the effects of it was not. […] The biggest surprise, I think, was that it was noticeable at all. I did not expect a seven percent bodyweight gain to have such noticeable effects on me.*
(Kevin, 28)

For some, these changes were accompanied by decreased self-confidence: 


*[I felt] A bit less confident to be honest, a bit quieter in social situations ‘cause I always felt like if I’m doing something like not really loud and daft, I’m just like, ‘I can’t over extend, because I’ve got so much weight here and all that’.*
(Marcus, 22)

These changes, in turn, influenced some men’s motivation or desire to participate in previously valued activities, including physical activity/exercise: 


*It’s probably the first time I’ve felt kind of self-conscious in any way about how I looked. […] It was like a twang of self-conscious […] if I had put on a lot more weight, I would definitely… I could see how this would lead me to kind of not doing those kind of things, or avoiding those kind of activities [i.e., frequenting gym/swimming pool].*
(Philip, 28)

## 4. Discussion

This qualitative study focused on how young adult men perceive their body image, their expectations and experiences of purposively gaining weight via increased food intake and what these reveal about broader sociocultural meanings around food, consumption and male body image. These findings are important as they provide a novel insight into young men’s experiences of changes in their body composition and eating practices which are not well understood, and the sociocultural influences, including the sometimes-flippant valorisation of ‘junk food’ and unhealthy eating among men of this age. 

The findings reveal how young adult men’s eating practices are congruent with common cultural understandings of masculine practice in the UK, specifically the notion that consuming large quantities of food is often synonymous with performances of masculinity [7,36,37]. Lupton and Feldman [38] delineate how excessive consumption of processed foods (high in sugar and fat) are endorsed due to their perceived cultural status, specifically across popular social media channels (such as YouTube) and imply that such performances of masculinity are congruent with ‘lad culture’ (e.g., consuming alcohol, ‘having a laugh’ and employing ‘banter’ etc) [39]. The results of the current study are consistent with these ideas, particularly the men’s use of language (e.g., ‘fun’, ‘laugh’ or ‘game’) in regard to being granted the opportunity to consume unlimited quantities of energy dense/unhealthy foods during the weight-gain phase. Moreover, several of the men described others’ reactions to their attempts to gain weight and being encouraged to ‘pig out’ as a source of humour. 

The findings are also consistent with previous research suggesting that some men may have a tendency to demonstrate a relative lack of concern in relation to overeating and/or may experience greater encouragement to overconsume [24] in comparison with women. Overall, these findings illustrate the ways in which ‘laddism’ and cultural constructions of masculinity reinforce attitudes to food and young men’s consumption of food. 

While most of the men expressed being generally satisfied with their body weight before taking part in the weight-gain component of the GlasVEGAS study, almost all articulated at least some form of dissatisfaction with their physique, usually a desire to be more muscular and/or leaner. Further, while some, particularly younger men, had attempted to lose weight (body fat) previously, others described explicitly having attempted to increase their muscle size by ‘bulking up’ and some said they had initially been motivated to take part in GlasVEGAS as they anticipated they may look slightly ‘healthier’ as a result of gaining weight. These findings are consistent with suggestions that body dissatisfaction in young men has increased [18,28] and there may be increased pressures for males to orient towards attaining a specific idealised physical shape [13]. These physical ideals have been identified as being espoused by both mass media and popular culture [22] including social media [11,40,41] and reality television [42] and may consequently enhance male body dissatisfaction [43], which has been associated with adverse behaviours (e.g., eating disorders, excessive exercise and steroid use) and mental health issues [44,45]. While some other men viewed attempts to alter their body shape or build as being ‘dangerous’, ‘superficial’ or ‘not worth time’, especially older men. These findings are congruent with other research showing that men may construct body dissatisfaction as a feature of adolescence, in combination with an absence of self-care or an inability to capitalise on other domains (e.g., occupation success or parenting status) to enhance individual masculine ‘capital’ [31]. 

During the weight-gain phase, some men reported noticing minimal or no changes in their physical appearance or psychological wellbeing, despite gaining ~7% of their body weight. While the current qualitative study was conducted within the context of the GlasVEGAS study, itself a controlled experiment, our findings demonstrate, for the first time, so far as we are aware, how significant weight-gain can occur rapidly over a relatively brief time and go unnoticed. 

Crucially, however, some men reported that they quickly embraced unhealthy eating practices, including developing cravings or a palate for processed foods, illuminating how relatively short term (i.e., 6 weeks) dietary changes may lead to the adoption of adverse eating habits in the longer term. Such changes might be more likely to occur during life course transitions common among young or emerging adults (e.g., entering higher education/employment, cohabiting with others or becoming parents themselves) [46,47]. This could help explain why this age-group may be vulnerable to rapid weight-gain [48]. Moreover, young adults have been shown to have poorer outcomes in weight management interventions compared to older adults [49].

A few men valued changes in their appearance associated with weight-gain, including appearing larger or having increased muscle size. These findings are consistent with research indicating that young men may be less concerned than young women about weight-gain [10]. They are also in line with previous research illuminating the complex pathways through which men may seek to enhance their muscle or physical size to accrue masculine ‘capital’ or ‘credit’ [5,50]. 

In contrast to those experiencing minimal, or even positive impacts associated with weight-gain, other men perceived both considerable physical changes, including increased body fat, decreased fitness levels and negative psychological consequences, such as reduced confidence or perceived attractiveness, and increased discomfort with their bodies. These results are consistent with the feelings of shame and embarrassment about their bodies and trepidation about receiving information from others confirming their weight status that were expressed by middle-aged men within the context of a weight-management programme [51]. Some men in the current study expressed reduced desire or motivation to engage in previously valued health-related behaviours, including physical activity, following weight-gain. The findings are consistent with theories of motivation and behaviour change, specifically the importance of ‘high quality’ motivation, illustrating how one can rapidly become de-motivated when lacking perceived competence or value in relation to performing behaviours important for sustaining health [52]. Hence, they provide valuable insight into the potential adverse impact of weight-gain on young men’s motivation for continued engagement for health enhancing activities.

Young adult men are especially underserved in existing weight management research [53,54]. Our study has identified several factors, including the valorisation of unhealthy foods, wider social influences on eating behaviour and male body image ideals, that are vital to consider when developing initiatives targeting young men in weight management, dietary or lifestyle interventions and public health promotion efforts.

### Limitations

While this study employed a longitudinal qualitative design to address the research aims, it is important to consider some limitations. The sample was fairly small and predominantly comprised of university students, hindering generalisability. In future it would be advantageous to consider including men from diverse social and economic backgrounds. However, recruitment to a study of intentional weight-gain raises ethical concerns in certain cases. The diversity of the sample was constrained for this study by the eligibility criteria for the GlasVEGAS study. The interviewer was a White Scottish male post-doctoral researcher in his thirties employed by the university where the main GlasVEGAS study was being conducted, and, given their relatively similar characteristics, the men in the qualitative study may have been able to relate closely to him. This may have allowed these men to provide open and frank accounts during their interviews, especially issues regarding body image and eating behaviour. However, while the men were invited to express their honest opinions during the interviews, and the independence of this qualitative work from the main GlasVEGAS study was emphasised, it is possible that some men may have felt inhibited from giving a more critical view of their experiences of taking part in the GlasVEGAS study more generally. 

## 5. Conclusions

The findings of the current study provide valuable insight on wider sociocultural meanings regarding cultures of food consumption and body image in relation to constructions of masculinities in young adult men. Most men in the study viewed the foods provided as part of the weight gain phase of the GlasVEGAS study as ‘treat(s)’ or ‘luxury’ items despite being of low nutritional value. The weight-gain process also prompted men to reflect on how cultural norms amplify overeating behaviours, particularly in social situations or environments involving family or friends. Several men described being surprised at how quickly they assimilated unhealthy eating habits and/or gained weight. Some valued changes in their appearance associated with weight-gain, including appearing larger or having increased muscle size. These findings suggest that young men may be particularly vulnerable to engaging in pursuits to enhance their body size that align with unrealistic standards and are potentially detrimental. These factors are vital to consider when developing weight management initiatives targeting young men, including the valorisation of unhealthy foods, wider social influences on diet and male body image ideals.

## Figures and Tables

**Table 1 ijerph-20-03320-t001:** Characteristics of the male study participants interviewed (*n* = 13).

Pseudonym	Age	Employment Status	Interview Time Point	Number of Interviews
Malcolm	20	Student	T1, T2	2
Philip	28	Student	T1, T2	2
Robert	25	Student	T1, T2	2
Davie	19	Student	T1, T2	2
Andy	30	Employed	T1, T2	2
Simon	21	Student	T1, T2	2
Luke	22	Student	T1, T2	2
Matt	21	Student	T1, T2	2
Douglas	19	Student	T1, T2	2
Jack	20	Student	T1, T2	2
Kieran	24	Student	T2 only	1
Marcus	22	Student	T2 only	1
Kevin	28	Student	T2 only	1

T1: at time of GlasVEGAS study baseline assessment; T2: at time of GlasVEGAS study weight gain follow-up assessment.

## Data Availability

The data are not suitable for sharing as contain information that could compromise the privacy of participants. The interview topic guide(s) are available in Appendix A.

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
