# Peer review of "A Qualitative Study on Young Men’s Experiences of Intentional Weight-Gain"

_ijerph, 2023, doi:10.3390/ijerph20043320_

Round 1

Reviewer 1 Report

General comments

-        I found this informative, well written and well designed.

-        I understand ethical approval was granted for the study, but can you explain a little about how these men were supported to go back to their eating practices after the study and can you address any criticism that the temporary weight gain was dangerous for their health?

-        Interview schedules seem thorough but well crafted with appropriate prompts, intonation & really good ‘anything else’ questions. Also commend interviewers for their manner in so professionally explaining purpose of the research in a really seemingly friendly way.

-        I have some small comments relating to specific sections below:

Abstract

-        Can you specify the time difference between baseline & follow up

Intro

-        Pg 2 This needs revision I think: “Consistent with the notion that gendered ideals 51 around masculinities and body image oscillate, depending on what is considered salient within a 52 particular sociocultural context [19], it has been suggested men are more likely to desire increased 53 body size in a ‘drive for muscularity’ [20] and inconsistent with what is considered, in biomedical 54 terms, to be ‘normal’ or ‘healthy’ [21] .” Issues: 1) Font change 2) end of section does not read fluently and 3) the desire to increase body size often isn’t in contrast to weight loss – the ideal is for men to gain defined and prominent muscle mass whilst losing body fat – to be mesomophic. I think this needs acknowledging. Some men may however ‘settle’ for increasing their overall size especially because/ if losing body fat is particularly hard.

-        Relatedly: “Findings suggest that men classed as both ‘underweight’ and ‘obese’ are more likely 56 to experience body dissatisfaction and weight stigma than those classified as ‘normal’ 57weight” you’ve omitted men who are classified as overweight. Some such men may be mesomorphic and I think BMI has been critiqued for failing to account for weight related to muscularity and weight related to body fat.

-        Font change from line 68-71 needs correction (and please correct throughout). Also this sentence reads less fluently than the one preceding it and might need to be integrated in the paragraph earlier.

-        Great explanation of double bind of masculinity

Method

-        Great to see a sample that isn’t predominantly White Europeans.

-        Some acknowledgement that South Asian people in the UK can have dual nationalities / identities or consider themselves solely British regardless of parentage or generational status would be good and would avoid any (racist) misinterpretations by others that ethnicity dictates nationality (or that Europe or the UK belongs to White people). I appreciate this is a little complicated as South Asian refers to nation states whilst also often being used as a proxy for ‘race’ (in the UK akin to White and Black) despite ‘race’ being  socially constructed. You’ve acknowledged this a little in noting that it was White Europeans who took part in your study (though your first mention of your other group is ‘European’). Just a little clarification and perhaps a disclaimer here I think would be really good.

-        Can you explain how the food was selected (especially dried fruit and nuts)?

-        Can you explain why 13 of the 21 men were approached and not all of the 21 ?

-        I’ve just read further and understood South Asian men weren’t included in the study. This is a shame. If you can make this clearer above (unless it was my error) and indicate whether South Asian men participated in the wider study that would be helpful and perhaps any efforts you took to recruit this under represented group (eg as supplemental information)

Results

-        Its really interesting to me how Luke refers to men including using “them”

-        Can you remind us what Andy’s age was when mentioning he was the oldest?

-        Interesting first theme, it echoes with other research I know and is well described I think. You show the range of responses to this whilst unpicking a central thread of contending with male appearance pressures.

-        Could you expand on participants pre emptying and justifying noticeable weight gain? I’d love to learn more.

-        Can you add in the 3.3. subheading this is from the follow up data and perhaps in the section before that this is baseline? 

Reviewer 2 Report

Although the article is very well structured, however, a finished process of updating its references must be carried out since more than 63% of the references are out of date. Therefore, this is a relevant aspect to consider in improving the article, since updating the theoretical framework and the discussion will greatly improve its quality.

There are other details to improve in the conclusion, it should only be limited to answering the objective of the study and making references and citations to other studies. This could go in the discussion or in a section on future projections, but not the conclusion.
